# Poly(l-Ornithine)-Based Polymeric Micelles as pH-Responsive Macromolecular Anticancer Agents

**DOI:** 10.3390/pharmaceutics15041307

**Published:** 2023-04-21

**Authors:** Miao Pan, Chao Lu, Wancong Zhang, Huan Huang, Xingyu Shi, Shijie Tang, Daojun Liu

**Affiliations:** 1Plastic Surgery Institute of Shantou University Medical College, Shantou 515041, China; 2Department of Plastic Surgery and Burn Center, Second Affiliated Hospital of Shantou University Medical College, Shantou 515041, China; 3Shantou Plastic Surgery Clinical Research Center, Shantou 515041, China; 4College of Pharmacy, Jinan University, Guangzhou 511436, China; 5Department of Pharmacy, Shantou University Medical College, Shantou 515041, China

**Keywords:** poly(l-ornithine), anticancer peptide, pH-responsive, membrane lysis

## Abstract

Anticancer peptides and polymers represent an emerging field of tumor treatment and can physically interact with tumor cells to address the problem of multidrug resistance. In the present study, poly(l-ornithine)-*b*-poly(l-phenylalanine) (PLO-*b*-PLF) block copolypeptides were prepared and evaluated as macromolecular anticancer agents. Amphiphilic PLO-*b*-PLF self-assembles into nanosized polymeric micelles in aqueous solution. Cationic PLO-*b*-PLF micelles interact steadily with the negatively charged surfaces of cancer cells via electrostatic interactions and kill the cancer cells via membrane lysis. To alleviate the cytotoxicity of PLO-*b*-PLF, 1,2-dicarboxylic-cyclohexene anhydride (DCA) was anchored to the side chains of PLO via an acid-labile *β*-amide bond to fabricate PLO(DCA)-*b*-PLF. Anionic PLO(DCA)-*b*-PLF showed negligible hemolysis and cytotoxicity under neutral physiological conditions but recovered cytotoxicity (anticancer activity) upon charge reversal in the weakly acidic microenvironment of the tumor. PLO-based polypeptides might have potential applications in the emerging field of drug-free tumor treatment.

## 1. Introduction

Even though early screening, advances in diagnostics, and improved therapeutic regimens have led to a decline in cancer mortality, malignancies remain the major cause of death [1]. Chemotherapy is one of the most important and indispensable methods for treating malignant tumors. However, chemotherapy is usually unsatisfactory due to insufficient drug accumulation in tumor tissues, poor aqueous solubility, drug resistance, severe off-target toxicity, and a high probability of metastasis [2,3,4,5]. To address these problems, a wide variety of polymeric micelles have been extensively explored to deliver anticancer drugs to the tumor site by embedding a drug in the hydrophobic core of the polymeric micelles or by conjugating the drug at the distal end, thus increasing the circulation time, improving the accumulation in tumor tissue, and reducing the toxicity of the drug itself [6,7,8,9,10]. Although significant advances in drug delivery systems have been achieved, there remain many challenges, such as burst release and potential off-target toxicity of small-molecule drugs, as well as their susceptibility to developing drug resistance [11,12]. To this end, new anticancer agents that afford high selectivity toward cancer cells and overcome multidrug resistance are in critical demand.

Host defense peptides are short cationic peptides which are widely found in nature and play an important role in immediate nonspecific defenses against various microbes, including bacteria, fungi, protozoa, and viruses [13]. Over the past few decades, inspired by host defense peptides, synthetic antimicrobial peptides (AMPs) have been extensively studied to combat bacteria via a membrane-lytic mechanism [14,15,16,17,18,19]. Cationic and amphiphilic AMPs directly interact with anionic microbial membranes via electrostatic interactions and insert hydrophobic blocks into microbial membrane lipid domains, thus exhibiting high selectivity toward microbes versus normal mammalian cells. Similar to the outer surface of bacterial cell membranes, negatively charged components (phosphatidylserine, sialic acid residues, and heparin sulfate) also reside on the outer membrane leaflet of cancer cells, leading to a relatively negatively charged cell surface [20,21,22]. Meanwhile, the loss of cholesterol enhances the fluidity of the cancer cell’s outer membrane [23]. Therefore, it is expected that cationic AMPs may selectively bind to the negatively charged membranes of cancer cells, destabilize the cell membrane, and lead to death.

The antibacterial activity of AMPs is closely related to their specific structures, such as the sequence of amino acids, structural folding, net charge, hydrophobic/hydrophilic balance, and molecular weight [24,25,26,27,28]. Earlier studies have shown that AMPs and antimicrobial polymers can potentially be utilized for the treatment of cancer [29,30]. Liu et al. presented anticancer heterochiral *β*-peptide polymers to combat multidrug-resistant cancers. The optimal polymer showed superior stability against proteolysis, low cost, and potent and broad-spectrum anticancer activities against multidrug-resistant cancer cells via a membrane-damaging mechanism [31]. Chen et al. synthesized a cationic anticancer polypeptide through the ring-opening polymerization of γ-allyl-l-glutamate-*N*-carboxyl anhydride and a subsequent thiol-ene click modification with cysteamine, which exhibited broad spectrum anticancer activity and directly induced rapid necrosis of cancer cells through a membrane-lytic mechanism [32]. Jan et al. designed cationic one-dimensional fibril assemblies formed from coil-sheet poly(l-lysine)-*b*-poly(l-threonine) block copolypeptides for cancer therapy [33], and Liu et al. reported a polymeric carrier for AMPs that could finely control the spatial distribution of AMPs in different biological microenvironments, thereby effectively enhancing their anticancer efficacy while minimizing their potential side effects [34]. Yang et al. reported on the use of a series of quaternary ammonium-based cationic macromolecules as chemotherapeutic agents to address drug resistance, where the cationic polymer selectively bound and lysed the cancer cell membrane [35].

In our previous research, unnatural amino acid-based star-shaped poly(l-ornithine)s, as AMPs, showed remarkable proteolytic stability, excellent biofilm-disrupting capacity, and broad-spectrum antimicrobial activity [36]. In the present study, amphiphilic block copolypeptides of poly(l-ornithine)-*b*-poly(l-phenylalanine) (PLO-*b*-PLF) were employed to construct polymeric micelles, aiming to explore the potential anticancer activity of unnatural poly(l-ornithine)s. PLO-*b*-PLF copolypeptides self-assemble into nanosized polymeric micelles with peripheral PLO arms that are able to disrupt negatively charged cancer cell membranes via electrostatic interactions, leading to membrane lysis. To alleviate the cytotoxicity of cationic PLO chains toward mammalian cells, 1,2-dicarboxylic-cyclohexene anhydride (DCA) was employed to modify the side chains of PLO to form the charge-reversal derivative PLO(DCA)-*b*-PLF [37,38]. PLO(DCA)-*b*-PLF showed negligible hemolysis and cytotoxicity to mammalian cells under neutral physiological conditions, but it also showed anticancer activity as a result of negative-to-positive charge conversion in the weakly acidic microenvironment of the tumor (pH 6.5–6.8) [39,40,41].

## 2. Materials and Methods

### 2.1. Synthesis and Characterization of Polypeptides

Block polypeptides PLO-*b*-PLF and PLO(DCA)-*b*-PLF were synthesized following the route depicted in Figure 1. Hexylamine was employed to initiate the sequential ring-opening polymerization of δ-benzyloxycarbonyl-l-ornithine *N*-carboxyanhydride (ZLO-NCA) and l-phenylalanine *N*-carboxyanhydride (LF-NCA) to produce poly(δ-benzyloxycarbonyl-l-ornithine)-*b*-poly(l-phenylalanine) (PZLO-*b*-PLF), followed by the deprotection of benzyloxycarbonyl groups of PZLO according to our previous report [36] to produce the final product PLO-*b*-PLF. PLO-*b*-PLF was modified with DCA to prepare PLO(DCA)-*b*-PLF. The detailed methods of synthesis and characterization are described in the Appendix A section.

### 2.2. Preparation and Characterization of Polymeric Micelles

PLO-*b*-PLF and PLO(DCA)-*b*-PLF were dissolved in PBS (0.01 M, pH 7.4) at a concentration of 1 mg/mL, followed by ultrasonication for 5 min. The particle size, polydispersity, and zeta potential of the polymeric micelles were determined by dynamic light scattering (DTS Zetasizer Nano, Malvern Instruments, Worcestershire, UK). The measurements were carried out for 3 runs per sample, and the results are presented as the mean ± standard deviation. The critical micelle concentration (CMC) of PLO-*b*-PLF in PBS (pH 7.4) or PLO(DCA)-*b*-PLF in bicarbonate buffer (pH 9.2) was determined by fluorescence spectroscopy using pyrene as a probe [34]. Briefly, polypeptide solutions with varying concentrations in the range of 0.015–2000 μg/mL were incubated with pyrene (6.16 × 10^−7^ M) overnight at room temperature in the dark. The excitation spectra of these solutions were scanned from 300 to 360 nm at an emission wavelength of 395 nm using a fluorescence spectrometer (Horiba FluoroMax, Kyoto, Japan). The intensity ratios of from I_339.0_ to I_334.0_ were drawn as a function of the logarithm of polymer concentrations. Several drops of the polymeric micellar solution were placed on a carbon-coated 200 mesh copper grid and kept overnight at room temperature. The morphology of the polymeric micelles was then examined on a transmission electron microscope (TEM, JEM-F200, Tokyo, Japan) with an acceleration voltage of 200 kV.

### 2.3. Cell Culture

Cancer cells (HepG2, MCF-7, A549, BT474, HeLa, and MCF-7/ADR) and normal cells (HK-2 and LO2) were cultured in DMEM containing 10% fetal bovine serum and 1% penicillin/streptomycin under 90% humidity and 5% CO_2_ at 37 °C.

### 2.4. Cell Viability Assay

The cytotoxicity of PLO-*b*-PLF and PLO(DCA)-*b*-PLF was evaluated by an alamarBlue assay [42]. All cells were seeded onto 96-well plates at a density of 6 × 10^3^ cells in 100 μL of DMEM with 10% fetal bovine serum per well. After culturing for 24 h, the medium was replaced with fresh complete medium containing different concentrations of polypeptide ranging from 500 to 1.0 μg/mL. Wells without polypeptide treatment and without cells were set as the positive control and negative control, respectively. After incubation for 24 h, the medium was replaced with fresh complete medium containing 10% alamarBlue solution. Upon additional incubation for 2.5 h, the fluorescence intensity of each well was measured on an Infinite M200 microplate reader (Tecan, Zurich, Switzerland) at an excitation wavelength of 555 nm and emission wavelength of 590 nm. Cell viability was calculated by the following formula:Cell viability (%) = [(Fluorescence_polypeptide_ − Fluorescence_negative control_)/(Fluorescence_positive control_ − Fluorescence_negative control_)] × 100%

### 2.5. Hemolysis Assay

Fresh rat red blood cells were washed three times by suspending cells in PBS (pH 7.4) and then centrifuged at 3500 rpm for 10 min at 4 °C. The supernatant was removed, and the red blood cells were suspended in PBS (5%, *v*/*v*). Then, 50 μL of red blood cell suspension was added to 50 μL of PBS solution containing PLO-*b*-PLF or PLO(DCA)-*b*-PLF at various concentrations ranging from 62 to 8000 μg/mL in a 96-well microplate. The mixture was incubated at 37 °C for 1 h. The plate was then centrifuged at 3500 rpm for 10 min. Aliquots (30 μL) of the supernatant were transferred into the well of a 96-well microplate containing 70 μL PBS, and the absorbance was measured at 540 nm using an Infinite M200 microplate reader (Tecan, Zurich, Switzerland). The untreated blood cell suspension in PBS was used as the negative control, and a solution containing red blood cells lysed with 2% Triton X-100 was employed as the positive control. Each test was performed in three replicates. The percentage of hemolysis was calculated by the following formula:Hemolysis (%) = [(absorbance_sample_ − absorbance_PBS_)/(absorbance_Triton_ − absorbance_PBS_)] × 100%

### 2.6. Dead/Live Cell Staining

HepG2 cells were seeded onto 6-well plates at a density of 1.2 × 10^5^ cells per well and cultured in 2 mL of complete DMEM at 37 °C for 24 h. Then, fresh complete DMEM containing predetermined concentrations of PLO(DCA)-*b*-PLF (0 × IC_50_, 0.5 × IC_50_, IC_50_, and 2 × IC_50_) at pH 6.5 and 100 μg/mL PLO(DCA)-*b*-PLF at pH 7.4 was added and incubated for 24 h at 37 °C. The cells were washed with PBS (pH 7.4) three times and subsequently costained with calcein acetoxymethyl ester (calcein AM, 6 μM) and propidium iodide (PI, 2 μM) for 15 min at 37 °C. Finally, the cells were washed with PBS twice and imaged by fluorescence microscopy (Observer A1, Zeiss Merlin, Baden-Wuerttemberg, Germany). The excitation and emission wavelengths for calcein AM were 488 nm and 520 nm, respectively, while the excitation and emission wavelengths for PI were 530 nm and 620 nm, respectively.

### 2.7. Zeta Potential Measurement

HepG2 and HK-2 cells were plated in 6-well plates (5 × 10^5^ cells/well) and cultured in complete DMEM for 24 h. The cells were harvested and resuspended in complete DMEM at pH 7.4 containing predetermined concentrations of PLO-*b*-PLF or PLO(DCA)-*b*-PLF. After incubation for 30 min at 37 °C, the cells were centrifuged and resuspended in 1 mL of H_2_O. The zeta potential of the cells was measured using dynamic light scattering (DTS Zetasizer Nano, Malvern Instruments, Worcestershire, UK). Measurements were carried out at 3 runs per sample, and the results are presented as the mean ± standard deviation.

### 2.8. Lactate Dehydrogenase (LDH) Leakage Assay

HepG2 cells were seeded in a 96-well plate at a density of 6 × 10^3^ cells per well in 100 μL of complete DMEM and cultured for 24 h. The cells were then incubated with different concentrations of PLO-*b*-PLF in fresh complete medium for 45 min. Untreated cells were employed as a negative control for background LDH release, while cells treated with lysis buffer were set as the positive control for maximal LDH release. Subsequently, the 96-well plate was centrifuged at 3000 rpm for 5 min, and LDH release was determined according to the protocol provided by the supplier. Briefly, 50 μL of supernatant was incubated with 50 μL of working solution for 30 min, followed by the addition of 50 μL of stop solution. Absorbance at 490 nm was recorded on an Infinite M200 microplate reader (Tecan, Zurich, Switzerland). The percentage of LDH release was calculated by the following equation:LDH release (%) = [(absorbance_sample_ − absorbance_negative control_)/(absorbance_positive control_ − absorbance_negative control_)] × 100%

### 2.9. Flow Cytometry Study

HepG2 cells were seeded onto 6-well plates at a density of 1.5 × 10^5^ cells per well and cultured in 2 mL of complete medium at 37 °C for 24 h. Then, the medium was replaced with fresh complete medium containing predetermined concentrations (0 × IC_50_, 0.25 × IC_50_, 0.5 × IC_50_, IC_50_, and 2 × IC_50_) of PLO-*b*-PLF. After incubation for 60 min, the cells were washed with PBS, detached with trypsin, and centrifuged to discard the supernatant. Next, the harvested cells were subsequently suspended in the binding buffer and stained with an Annexin V-FITC Apoptosis Detection kit according to the protocol provided by the supplier. Finally, the Annexin V-FITC/PI-labeled cells were subjected to measurement on a BD Accuri™ C6 Flow Cytometer (Becton, Dickinson, and Company, Franklin Lakes, NJ, USA).

### 2.10. Morphological Visualization of Cancer Cells by Scanning Electron Microscopy (SEM)

HepG2 cells were seeded at a density of 1.2 × 10^5^/well on 10 mm × 10 mm sterilized coverslips in a 6-well plate and cultured in complete medium for 24 h, followed by incubation with predetermined concentrations (0 × IC_50_, 0.25 × IC_50_, 0.5 × IC_50_, IC_50_, 2 × IC_50_, and 4 × IC_50_) of PLO-*b*-PLF in fresh complete medium for 60 min at 37 °C. The cells were then washed with PBS twice and fixed with 2.5% glutaraldehyde solution at 4 °C overnight. Subsequently, the medium was removed, and the cells were gradually dehydrated by serial incubation in 30%, 50%, 70%, 85%, 95%, and 100% ethanol solutions. Finally, the cells on coverslips were visualized by SEM (Gemini 300, Zeiss Merlin, Baden-Wuerttemberg, Germany).

### 2.11. Confocal Laser Scanning Microscopy (CLSM) Study

HepG2 cells were seeded on sterilized coverslips in a 6-well plate at a density of 1.2 × 10^5^ cells per well in 2.0 mL complete medium and cultured for 24 h, followed by incubation with predetermined concentrations (0 × IC_50_, 0.5 × IC_50_, IC_50_, and 2 × IC_50_) of PLO-*b*-PLF in fresh complete medium for 30 min at 37 °C. The cells were then rinsed with PBS three times and fixed with 4% paraformaldehyde for 20 min. Subsequently, the cells were washed with PBS three times and then sequentially incubated with DiO (2 μM) for 15 min and Hoechst (10 μg/mL, 1 mL) for 5 min to stain the cell membranes and cell nuclei, respectively. After washing with PBS three times, the coverslips were observed with a CLSM microscope (Carl Zeiss, LSM 800, Baden-Wuerttemberg, Germany).

### 2.12. In Vitro Cancer Cell Migration Assay

HepG2 cells were seeded in 6-well plates at a density of 1.5 × 10^5^ cells per well. After incubation for 24 h at 37 °C, the plate surface was scratched with a 200 μL pipette tip to draw a gap of uniform width. The medium was replaced with fresh medium without fetal bovine serum and containing different concentrations (0.25 × IC_50_, 0.5 × IC_50_, IC_50_, and 2 × IC_50_) of PLO(DCA)-*b*-PLF at pH 6.5 or 100 μg/mL PLO(DCA)-*b*-PLF at pH 7.4. The cells were then incubated at 37 °C. Images of the gaps were taken using a bright field microscope (Observer A1, Zeiss Merlin, Baden-Wuerttemberg, Germany) at 0 h and 24 h after scratching.

## 3. Results and Discussion

### 3.1. Synthesis and Characterization of PLO-b-PLF and PLO(DCA)-b-PLF

The block polymer PLO-*b*-PLF consists of poly(l-ornithine) as the hydrophilic block and poly(l-phenylalanine) as the hydrophobic segment. The synthetic route of PLO-*b*-PLF is depicted in Figure 1, and the detailed procedures are described in the Appendix A section. In the present study, the degree of polymerization (DP) of PLO was fixed at 30, while the DPs of PLF were set at 0, 4, 8, and 12. Hexylamine was employed to initiate the sequential ring-opening polymerization of ZLO-NCA and LF-NCA to produce PZLO-*b*-PLF (i.e., PZLO_30_ (**PZ1**), PZLO_30_-*b*-PLF_4_ (**PZ2**), PZLO_30_-*b*-PLF_8_ (**PZ3**), and PZLO_30_-*b*-PLF_12_ (**PZ4**)), followed by the deprotection of the benzyloxycarbonyl groups of PZLO according to our previous report [36] to produce the final products PLO_30_ (**P1**), PLO_30_-*b*-PLF_4_ (**P2**), PLO_30_-*b*-PLF_8_ (**P3**), and PLO_30_-*b*-PLF_12_ (**P4**). The intermediate PZLO-*b*-PLF and the final product PLO-*b*-PLF were characterized using nuclear magnetic resonance (^1^H NMR) and gel permeation chromatography (GPC). A typical ^1^H NMR spectrum and GPC chromatograms of the PZLO-*b*-PLF intermediate are shown in Appendix A, and the calculated results are listed in Appendix A. Based on the ^1^H NMR spectra, the DPs of PZLO and PLF in **PZ1**, **PZ2**, **PZ3**, and **PZ4** were estimated to be 30.4 and 0.0, 30.4 and 4.09, 29.2 and 7.91, and 29.1 and 11.3, respectively (Appendix A). All of the calculated DP values are very close to the respective feeding ratios. The unimodal GPC peaks and their narrow distributions in Appendix A indicate that the synthesis of PZLO-*b*-PLF proceeded in a controlled manner. Due to the self-assembly of PLO-*b*-PLF in aqueous solution, the characterization of PLO-*b*-PLF with ^1^H NMR and GPC is difficult. For example, the GPC chromatograms of PLO-*b*-PLF show predominant peaks, exhibiting very large molecular weights, which can be ascribed to the self-assembled polymeric micelles (Appendix A). Consistently, the phenyl groups demonstrated very weak signals in the ^1^H NMR spectrum of PLO-*b*-PLF due to the aggregation of hydrophobic PLF blocks (Appendix A). However, it should be noted that the deprotection procedures also proceeded in a controlled manner according to our previous report [43,44].

To reduce cytotoxicity, the cationic PLO-*b*-PLF was modified with DCA to prepare anionic PLO(DCA)-*b*-PLF, i.e., PLO_30_(DCA) (**PD1**), PLO_30_(DCA)-*b*-PLF_4_ (**PD2**), PLO_30_(DCA)-*b*-PLF_8_ (**PD3**), and PLO_30_(DCA)-*b*-PLF_12_ (**PD4**), where DCA was anchored to the amino groups of the side chains of l-ornithine in PLO-*b*-PLF via acid-labile *β*-carboxylic amide linkages (Figure 1). A typical ^1^H NMR spectrum of PLO(DCA)-*b*-PLF is shown in Appendix A. The shift of the proton peak of *g* from 2.8 ppm to 3.1 ppm and the appearance of proton peak *h* at approximately 2.0 ppm indicated the successful modification of the DCA.

### 3.2. Characterization of PLO-b-PLF and PLO(DCA)-b-PLF Micelles

The amphiphilic block copolypeptides PLO-*b*-PLF and PLO(DCA)-*b*-PLF are expected to self-assemble into polymeric micelles in aqueous solution. The aggregation behavior, i.e., CMC, was investigated by fluorescence spectroscopy using pyrene as the probe according to a previously reported method [35]. The CMCs of cationic **P2**, **P3**, and **P4** were determined to be 179.5 μg/mL, 51.3 μg/mL, and 28.8 μg/mL, while the CMCs of anionic **PD2**, **PD3**, and **PD4** were calculated to be 162.2 μg/mL, 40.7 μg/mL, and 22.4 μg/mL, respectively (Appendix A). An elongation of the hydrophobic PLF block led to a prominent decrease in the CMC values, and the small CMC values indicated that the polymeric micelles, self-assembled from PLO-*b*-PLF and PLO(DCA)-*b*-PLF (especially **P4** and **PD4**), had high thermodynamic stability even when diluted in the blood stream.

Dynamic light scattering was employed to investigate the sizes and zeta potentials of polymeric micelles. As shown in Figure 1a, the hydrodynamic diameters of the polymeric micelles ranged from 52 nm to 113 nm. Elongation of the PLF block increases the particle size, whereas DCA modification slightly decreases the micellar diameter. The zeta potentials of the **P2**, **P3**, and **P4** polymeric micelles are positive, ranging between 14.6 mV and 20.4 mV (Figure 1b), which can be ascribed to the protonated state of amino groups along the l-ornithine side chains under physiological conditions. Modification with DCA converted the charging state from positive to negative (Figure 1b), which is indicative of the success of the DCA modification. The morphology of the polymeric micelles was examined by TEM (Appendix A). The polymers **P4** and **PD4** adopted a spherical morphology with a relatively uniform particle size [35]. The smaller sizes of polymeric micelles obtained from the TEM measurements might be ascribed to the collapse and contraction of micellar structures during the process of sample preparation.

### 3.3. Hydrolysis of Acid-Labile PLO(DCA)-b-PLF

In PLO(DCA)-*b*-PLF, the DCA groups were anchored to the side chains of PLO with acid-labile *β*-carboxylic amide linkages. It is expected that the *β*-carboxylic amide would remain relatively stable under physiological conditions (pH 7.4), enabling a negative-to-positive charge reversal to occur upon the hydrolysis of the *β*-carboxylic amide under the slightly acidic conditions of the tumor microenvironment (pH 6.5–6.8). The hydrolysis kinetics of PLO(DCA)-*b*-PLF were investigated by ^1^H NMR spectra and zeta potential measurements. The evolution of the ^1^H NMR spectra of PLO(DCA)-*b*-PLF as a function of incubation time at pH 6.5 is shown in Appendix A, in which the methylene peak of the ornithine side chains shifts from 3.1 ppm to 2.8 ppm as the *β*-carboxylic amide hydrolyzes. Two peaks (H_a_ and H_b_) were integrated to obtain the hydrolysis kinetics of PLO(DCA)-*b*-PLF under different pH values (Appendix A). The results showed that the *β*-carboxylic amide in PLO(DCA)-*b*-PLF is easily hydrolyzed at pH 6.5 but remains relatively stable at pH 7.4, thus realizing the negative-to-positive charge conversion required for antitumor activity. Meanwhile, the zeta potentials of **PD4** as a function of incubation time at pH 6.5 and 7.4 were measured (Figure 2). The surface charge of **PD4** was converted from negative to positive upon incubation at pH 6.5 after approximately 6.5 h, whereas **PD4** remained negatively charged even after 48 h of incubation at pH 7.4. These results indicate that negative-to-positive charge reversal of PLO(DCA)-*b*-PLF could be achieved by the pH-induced hydrolysis of *β*-carboxylic amide under the slightly acidic conditions of the tumor microenvironment.

### 3.4. Cytotoxicity (Antitumor Activity) Assays

The cytotoxicity of PLO-*b*-PLF block copolymers was evaluated against six cancer cell lines by alamarBlue assay, and the corresponding IC_50_ values are listed in Table 1. PLO-*b*-PLF exhibited a broad spectrum of anticancer activity against all cancer cells, including MCF-7/ADR cells, with IC_50_ values ranging from 5.03 to 20.3 μg/mL. Notably, the IC_50_ value of **P1** (in the absence of hydrophobic block and micelle formation) was close to those of **P2**, **P3**, and **P4**, bearing hydrophobic PLF segments, indicating that the presence of a hydrophobic segment and the increase in charge density as a result of micelle formation did not significantly improve the anticancer activity of PLO-*b*-PLF. Meanwhile, we measured the zeta potential of cancer cells, which resided at a very narrow range at about −35 mV (HepG2, −35.0 mV; MCF-7, −33.5 mV; Hela, −35.1 mV; A547, −35.8 mV). No obvious correlation between the IC_50_ values and zeta potentials of the cells was observed. The IC_50_ value might relate to the zeta potential of the cell as well as other factors, such as the cell membrane structure. Since **P4** had a lower CMC value and, thus, higher hydrodynamic stability that might exhibit better resistance to the dilution conditions in vivo, it was selected as the representative polymer for the following studies.

To alleviate the cytotoxicity of **PD4** toward normal cells, anionic and charge-reversal PLO(DCA)-b-PLF was synthesized. The cytotoxicity of **PD4** on representative HK-2 and LO2 cells at pH 7.4 is shown in Appendix A. The survival rates of HK-2 and LO2 cells were above 50%, even when treated at a dose of 500 μg/mL **PD4** for 24 h, which is indicative of a negligible toxicity of **PD4** under physiological conditions. Under the slightly acidic conditions of tumors, the hydrolysis of DCA could recover the cationic state and thus afford anticancer activity. The cytotoxicity of DCA-modified **PD4** on various cancer cells after incubation for 24 h at pH 6.5 was then determined (Appendix A). The corresponding IC_50_ values of **PD4** on HepG2, A549, MCF-7, MCF-7/ADR, BT474, and HeLa cells were 20.1 μg/mL, 28.7 μg/mL, 29.0 μg/mL, 12.9 μg/mL, 37.2 μg/mL, and 48.4 μg/mL, respectively. These results indicate that **PD4**, modified with acid-sensitive DCA, could effectively kill cancer cells in weakly acidic cancer tissue without causing harm to normal cells.

The anticancer effect of **PD4** was directly demonstrated through a live/dead cell costaining measurement. HepG2 cells were incubated in the presence of **PD4** under different pH values (6.5 or 7.4) and then stained with PI (red, dead cells) and calcein AM (green, live cells). As shown in Figure 3, few PI-positive cells were observed when treated with 100 μg/mL **PD4** at pH 7.4. However, when the concentration of **PD4** increased from 0.5 × IC_50_ to 2 × IC_50_ at pH 6.5, stronger PI staining was detected, indicating that more HepG2 cells were killed with increasing concentrations of **PD4**. The live/dead staining results are consistent with the alamarBlue assay and further demonstrate that the noncytotoxic **PD4** could experience charge reversal at pH 6.5 to exert anticancer activity.

### 3.5. Hemolysis Assay

The hemolytic activities of **P4** and **PD4** were determined against rat red blood cells, and the corresponding hemolytic percentage as a function of polypeptide concentrations is shown in Appendix A. For **P4**, hemolysis increased with increasing concentrations of **P4**. However, the hemolysis caused by **PD4** remained at approximately zero even when the concentration of **PD4** was as high as 4000 μg/mL. These results show that DCA modification can greatly reduce hemolytic toxicity, largely owing to the negatively charged state of **PD4**.

### 3.6. Anticancer Mechanism

#### 3.6.1. Electrostatic Binding of PLO onto the Surface of Cancer Cells

The binding of cationic PLO-based polypeptides onto the surface of anionic cancer cells was evaluated by measuring the change in zeta potential of HepG2 cells as a function of different concentrations of **P1** and **P4** [45]. HepG2 cells were incubated with **P1** or **P4** for 30 min, centrifuged, and then resuspended prior to zeta potential measurements. The surface charge of HepG2 cells evolved from negative to positive with increasing concentrations of **P1** and **P4** (Figure 4A), indicating that cationic PLO-based polypeptides bind to the surface of HepG2 cells via electrostatic interactions. It is interesting to note that, compared to **P4**, having micelle-forming characteristics, **P1** showed faster and greater HepG2 cell-binding capacity, plausibly owing to its flexible molecular conformation and higher l-ornithine content. In contrast, the surface charge of normal HK-2 cells treated with anionic **PD4** remained essentially constant with an increasing **PD4** concentration, suggesting a low affinity of **PD4** toward normal cells. These results revealed that electrostatic interactions between PLO-based polypeptides and cancer cells were responsible for their binding and represented the first step of anticancer action.

#### 3.6.2. Effect of PLO-Based Polypeptides on the Membrane Permeability of Cancer Cells

Once bound to cancer cells, AMPs are expected to destroy the integrity of the cell membrane, thus killing cancer cells via physical action [26]. At this stage, the leakage of the cytoplasmic enzyme LDH from HepG2 cells upon incubation with **P1** or **P4** was detected. As shown in Figure 4B, a higher extracellular content of LDH was detected with the increase in the concentration of **P1** or **P4**, indicating greater cell membrane damage. In addition, cells treated with **P1** (in the absence of the hydrophobic segment) exhibited greater LDH leakage, suggesting that the presence of a hydrophobic segment did not significantly improve the membrane-disrupting ability of PLO-based polypeptides. These results are consistent with the IC_50_ values and further show that PLO-based polypeptides exert their anticancer action via a mechanism involving physical membrane disruption (Figure 4C).

#### 3.6.3. Flow Cytometry Study

The Annexin V-FITC/PI apoptosis detection assay was used to investigate the anticancer mechanisms of the representative copolypeptide **P4**. Annexin V not only binds to the valgus phosphatidylserine which is exposed on the membrane outer leaflet of apoptotic cells, but also enters necrotic cells which are lacking membrane integrity and binds to phosphatidylserine on the inner leaflet of the bilayer [32]. In contrast, PI can only label the DNA of necrotic cells. As shown in Figure 5, compared to the control group, the proportion of necrotic cells was obviously improved with the increase in the concentration of **P4**. However, the percentage of apoptotic cells remained less than 5% in all **P4** treatment groups. The apoptosis results implied that **P4** damaged the cell membrane without obvious cell apoptosis.

#### 3.6.4. Cell Membrane Disruption Viewed by SEM and CLSM

The morphology and structural changes of cancer cells upon incubation with different concentrations of **P4** were further examined by SEM. HepG2 cells without PLO-*b*-PLF treatment showed a smooth and intact membrane surface (Figure 6). Upon treatment with **P4** at lower concentrations (0.25 × IC_50_ and 0.5 × IC_50_), the cell membrane surface became rough but remained intact. However, when the concentration of **P4** was further increased (IC_50_, 2 × IC_50_, and 4 × IC_50_), the cell membrane surface became rougher. Eventually, cell morphology was gradually lost.

The polypeptide-induced cell membrane disruption was further investigated by CLSM. DiO and Hoechst were employed as membrane-staining and nucleus-staining dyes, respectively. As shown in Figure 7, **P4** showed dose-dependent cell membrane disruption. HepG2 cells treated with higher concentrations of **P4** exhibited a stronger DiO signal, indicating that more dye resided in the cytoplasm due to cell membrane damage. In other words, higher concentrations of **P4** led to more obvious cell membrane penetration. The CLSM results were consistent with the SEM results, further suggesting that **P4** kills cancer cells through membrane lysis.

### 3.7. Migration Inhibition

Inhibiting cancer cell migration represents an anticancer activity of chemotherapeutics. The ability of **PD4** to inhibit cancer cell migration at different pH values was evaluated using a scratch wound healing assay (Appendix A). After 24 h of incubation, cells treated with **PD4** at a dose of 0.25 × IC_50_ and pH 6.5 for 24 h showed reduced cell migration (Appendix A) compared to the untreated cells (Appendix A). Moreover, cells that were treated with 100 μg/mL **PD4** at pH 7.4 experienced partial migration (Appendix A). These results demonstrate the importance of acidic conditions on the anticancer effect of **PD4** and imply that **PD4** can not only kill cancer cells but also prevent their metastasis.

## 4. Conclusions

Cationic PLO-*b*-PLF and anionic PLO(DCA)-*b*-PLF were synthesized, characterized, and evaluated as macromolecular anticancer agents. These amphiphilic block copolypeptides self-assemble into nanosized polymeric micelles in aqueous solution. PLO-*b*-PLF micelles bind to the surface of cancer cells via electrostatic interactions, disrupt the cancer cell membranes, and kill cancer cells via membrane lysis. This physical mechanism might afford PLO-*b*-PLF with broad-spectrum anticancer activity and alleviate the problem of drug resistance. The DCA modification of PLO-b-PLF, i.e., PLO(DCA)-*b*-PLF, prevents cytotoxicity and hemolytic activity under normal physiological conditions. However, the negative-to-positive charge reversal of PLO(DCA)-*b*-PLF as a result of the hydrolysis of the *β*-amide bond, under weakly acidic conditions restores cytotoxicity (anticancer activity), improves the anticancer selectivity against tumor cells. It should be noted that low extracellular pH is seen not only in tumors but also in inflammation; therefore, immune cell viability may be impaired. Although an enhanced permeation and retention effect is expected for nanosized PLO(DCA)-*b*-PLF micelles, the applicability of the pH-responsive antitumor activity of PLO(DCA)-*b*-PLF requires further careful evaluation. The in vivo antitumor evaluation of PLO(DCA)-*b*-PLF micelles is currently underway and will be reported in the future.

## Data Availability

Not applicable.

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
