# Peer review of "Poly(l-Ornithine)-Based Polymeric Micelles as pH-Responsive Macromolecular Anticancer Agents"

_pharmaceutics, 2023, doi:10.3390/pharmaceutics15041307_

Round 1

Reviewer 1 Report (Previous Reviewer 2)

Looks in GOOD form now. 

Minor modifications and spell checks are suggested. 

Author Response

Implemented. Spell checks have been carefully conducted and the manuscript has been modified accordingly in Line 39, Line 53, Line 157, Line 258, and Line 367.

Reviewer 2 Report (Previous Reviewer 3)

The authors took into account all my suggestions for the text of the manuscript.  Corrections to minor methodological errors and text editing are needed to the main text as well as the supplementary file. After improvement, this manuscript could be recommended for publication.

Author Response

Implemented. Minor methodological errors and text editing have been conducted, the revision has been highlighted in pink color in the main text and the Supplementary Materials.

Reviewer 3 Report (Previous Reviewer 4)

I thank the authors for addressing my comments and agree to the replies and the emanedments to the text. I did not find the amendments, however, they mentioned in the answer." The zeta potentials of cancer cells were measured and they resided at a very narrow range about -35 mV (HepG2, -35.0 mV; MCF-7, -33.5 mV; Hela, -35.1 mV; A547, -35.8 mV). No obvious correlation between IC50 values and zeta potentials of the cells was observed. The IC50 value might relate to the zeta potential of the cell as well as other factors such as the cell membrane structure." It would be good to include this also in the text and suggest the plasma membrane structure, they have in mind causing the differences between normal cells and cancer cells.

Author Response

Thanks for your suggestion. The amendments of “Meanwhile, we measured the zeta potential of cancer cells, which resided at a very narrow range of about -35 mV (HepG2, -35.0 mV; MCF-7, -33.5 mV; Hela, -35.1 mV; A547, -35.8 mV). No obvious correlation between IC50 values and zeta potentials of the cells was observed. The IC50 value might relate to the zeta potential of the cell as well as other factors, such as the cell membrane structure” have been incorporated in Line 321-326, Page 8 and highlighted in turquoise color.

This manuscript is a resubmission of an earlier submission. The following is a list of the peer review reports and author responses from that submission.

Round 1

Reviewer 1 Report

Miao Pan and colleagues present a quality and well-written experimental manuscript describing poly(L-ornithine)-based polymeric micelles as pH-responsive and anti-drug resistant macromolecular anticancer agents.

Authors prepared poly(L-ornithine)-b-poly(L-phenylalanine) (PLO-b-PLF) to block copolypeptides and evaluated as macromolecular anticancer agents. Amphiphilic PLO-b-PLF self-assembled into nanosized polymeric micelles in an aqueous solution. The cationic PLO-b-PLF micelles interacted steadily with the negatively-charged surfaces of cancer cells via electrostatic interactions and eventually kill the cancer cells via membrane lysis. In order to alleviate the cytotoxicity of PLO-b-PLF, 1,2-dicarboxylic-cyclohexene anhydride was anchored to the side chains of PLO via an acid-labile beta-amide bond to fabricate PLO(DCA)-b-PLF. Anionic PLO(DCA)-b-PLF showed negligible hemolysis and cytotoxicity under neutral physiological conditions but recovered cytotoxicity upon charge reversal in the weakly acidic micro-environment of the tumor.

Authors employed amphiphilic block copolypeptides of poly(L-ornithine)-b-poly(L-phenylalanine) (PLO-b-PLF) to construct polymeric micelles, aiming to explore the potential anticancer activity of unnatural poly(L-ornithine)s. PLO-b-PLF copolypeptides self-assemble into nanosized polymeric micelles with peripheral PLO arms able to disrupt negatively-charged cancer cell membranes via electrostatic interaction, leading to membrane lysis. In order to alleviate the cytotoxicity of cationic PLO chains towards mammalian cells, 1,2-dicarboxylic-cyclohexene anhydride (DCA) was employed to modify the side chains of PLO to form charge-reversal derivatives PLO(DCA)-b-PLF. PLO(DCA)-b-PLF showed negligible hemolysis and cytotoxicity to mammalian cells under neutral physiological conditions but anticancer activity as a result of negative-to-positive charge conversion in the weakly acidic micro-environment of the tumor (pH 6.5-6.8). They found that pH-responsive PLO(DCA)-b-PLF can selectively kill cancer cells by membrane lysis, thus avoiding the problem of drug-resistance.

Finally, authors conclude that amphiphilic block copolypeptides self-assemble into nanosized polymeric micelles in an aqueous solution. PLO-b-PLF micelles bind to the surface of cancer cells via electrostatic interactions, disrupt the cancer cell membranes, and kill cancer cells via membrane lysis. The physical mechanism affords PLO-b-PLF with broad-spectrum anticancer activity, including inhibition of drug-resistant cancer cells. The DCA modification of PLO-b-PLF, i.e., PLO(DCA)-b-PLF, preventscytotoxicity and hemolytic activity under normal physiological conditions. However, the negative-to-positive charge reversal of PLO(DCA)-b-PLF, as a result of the hydrolysis of betamide bond, under weakly acidic conditions restores cytotoxicity (anticancer activity), thus improving the anticancer selectivity against tumor cells. The nanometer diameter size of PLO(DCA)-b-PLF micelles might also offer enhanced permeation and retention into a tumor. The in vivo antitumor evaluation of PLO(DCA)-b-PLF micelles is currently underway and will be reported in the future.

Overall, the manuscript is highly valuable for the scientific community and should be accepted for publication.

===========

Other comments:

1) Please check for typos throughout the manuscript.

2) With regards to cytotoxicity evaluation in MCF7 cells  - authors are kindly encouraged to cite the following article that describes a relevant experimental protocol. DOI: 10.22099/mbrc.2019.34179.1419

Reviewer 2 Report

I have gone through the manuscript. Topic is indeed interesting but it needs extensive conditioning. The Authors have presented the potential of polymeric micelles as delivery vehicles. However, it needs to be tested in additional cancer cell lines. HepG2 cells are not tumorigenic in nude mice. Besides, many previous publications also demonstrated that HepG2 cells are nontumorigenic in nude mice. This discrepancy raised our concerns about the authentication of HepG2 cells in the paper. We do not know whether the identity of HepG2 cell line was confirmed by short tandem repeat profiling or any other methods.

Apoptosis needs to be tested in details. It can be tested atleast through analysis of caspases. 

Reviewer 3 Report

Draft with the title “Poly(L-ornithine)-based polymeric micelles as pH-responsive and anti-drug resistant macromolecular anticancer agents” describes studies in the field of anticancer drug design and presents interest for the modern pharmacy.

However, I have a number of comments on this draft:

Title – use of colocation “anti-drug resistant” is not correct, and moreover – non-relevant to the present research! The authors did not conduct or describe any research that would concern the study of this aspect.

Abstract

18 - The word “eventually” couldn’t be used in such experimental work! It’s not scientific at least!

Introduction

89-90 – The sentence “The pH-responsive PLO(DCA)-b-PLF can selectively kill cancer cells by membrane lysis, thus avoiding the problem of drug-resistance.” – It sounds like a preliminary conclusion for this work, however, what are there pieces of evidence or reasons to claim this? I haven't seen any after reading all the draft!

 Materials and Methods

101 – I haven’t found detailed information about synthesis in the “Supplementary Materials”. The same comment concerns row 217 in the “Results and discussion” section.

103 – I suggest placing Scheme 1 in the “Results and discussion” section.

 Results and discussion

245-246 - I suggest that the authors should pay more attention and present more information concerning data of the 1H NMR spectra, namely there is no information about the behavior/pattern of NH2- and NH-groups protons for the synthesized compounds

 357 – should the word “bacteria” be here?

 Conclusions

412 – 413 – The sentence “The nanometer diameter size of PLO(DCA)-b-PLF micelles might also offer enhanced permeation and retention into a tumor.” – it’s just an assumption, which wasn’t studied and confirmed in the present work, so it should be removed from the “Conclusions” section

File with “Supplementary Materials” – should be improved

So, my decision is that the draft in the present version needs major revision.

Reviewer 4 Report

Pan et al. characterized poly(L-ornithine)-based polymeric micelles attached to DCA to convert their cationic charge to anionic and thereby decrease cytotoxicity. By change of the pH from neutral to acidic, cytotoxicity would be restored. The manuscript is clearly written and the provided data, in general, support the theoretical concept.

Comments

- The sensitivity (IC50) should be correlated to the zeta potential of the cells to support the proof of concept

-What is the explanation for the reduced sensitivity of the MCF-7/ADR DOX resistant cells compared to the MCF-7 cells?

Minor

The following information should be added:

provider of the cell lines

if experiments were performed in absence or presence of FBS in the media

company of DTS or (DLS?) Zetasizer

company and location for the Tecan Infinite Pro series M200 microplate reader in the same way as the for the other instruments.

Additional Comments:

The characterized micelles are not completely new but an improvement compared to the existing ones because they are provided with tumor-targeting (low pH) capability. The addition of this functionality could result in improved cancer treatment, where there is always a need for improved therapies.

The link between membrane potential of the (tumor) cells and sensitivity to the micelles needs to be shown by correlation of the sensitivity by IC50 to the zeta potential of the cells.

It might be good to discuss the limitation of the concept that low extracellular pH is not seen only in tumors but also in inflammation and discuss if (given the correlation of cell zeta potential and cytotoxic effect has been shown) immune cell viability may be impaired.